# How Well Do Rodent Models of Parkinson’s Disease Recapitulate Early Non-Motor Phenotypes? A Systematic Review

**DOI:** 10.3390/biomedicines10123026

**Published:** 2022-11-24

**Authors:** Tracy D. Zhang, Scott C. Kolbe, Leah C. Beauchamp, Ella K. Woodbridge, David I. Finkelstein, Emma L. Burrows

**Affiliations:** 1Florey Institute of Neuroscience and Mental Health, Melbourne Brain Centre, University of Melbourne, 30 Royale Parade, Parkville, VIC 3052, Australia; 2Department of Neuroscience, Central Clinical School, Monash University, 99 Commercial Road, Melbourne, VIC 3004, Australia

**Keywords:** Parkinson’s disease, genetic rodent models, prodromal PD, non-motor PD phenotypes, systematic review

## Abstract

The prodromal phase of Parkinson’s disease (PD) is characterised by many non-motor symptoms, and these have recently been posited to be predictive of later diagnosis. Genetic rodent models can develop non-motor phenotypes, providing tools to identify mechanisms underlying the early development of PD. However, it is not yet clear how reproducible non-motor phenotypes are amongst genetic PD rodent models, whether phenotypes are age-dependent, and the translatability of these phenotypes has yet to be explored. A systematic literature search was conducted on studies using genetic PD rodent models to investigate non-motor phenotypes; cognition, anxiety/depressive-like behaviour, gastrointestinal (GI) function, olfaction, circadian rhythm, cardiovascular and urinary function. In total, 51 genetic models of PD across 150 studies were identified. We found outcomes of most phenotypes were inconclusive due to inadequate studies, assessment at different ages, or variation in experimental and environmental factors. GI dysfunction was the most reproducible phenotype across all genetic rodent models. The mouse model harbouring mutant A53T, and the wild-type hα-syn overexpression (OE) model recapitulated the majority of phenotypes, albeit did not reliably produce concurrent motor deficits and nigral cell loss. Furthermore, animal models displayed different phenotypic profiles, reflecting the distinct genetic risk factors and heterogeneity of disease mechanisms. Currently, the inconsistent phenotypes within rodent models pose a challenge in the translatability and usefulness for further biomechanistic investigations. This review highlights opportunities to improve phenotype reproducibility with an emphasis on phenotypic assay choice and robust experimental design.

## 1. Introduction

The Parkinson’s disease (PD) is the fastest-growing neurological disorder and affects over 6 million people globally as reported by the Global Burden of Disease in 2016 [1,2]. The incidence of PD is projected to double to 12.9 million by 2040 as the population ages [3]. The multitude of motor and non-motor symptoms associated with PD substantially affect quality of life and are poorly managed by current therapeutic approaches [4]. Unfortunately, there is no approved disease-modifying treatment for PD. One major hindrance in the development of novel treatments is the late clinical diagnosis rendering neuroprotective therapies ineffective. Current diagnosis relies on the development of hallmark motor symptoms of bradykinesia, rigidity, and tremors [5,6]. By the time of diagnosis, there is a 50–70% reduction in the dopaminergic cells in the substantia nigra pars compacta (SNpc) [7,8]. Additionally, within 5 years post-diagnosis, 50–90% of these cells have perished based on post-mortem studies [9]. This rate of cell loss remains relatively stable 27 years post-diagnosis, indicating that the most extensive neurodegenerative processes occur in the prodromal and early stages of clinical disease [9]. Adding complexity is that PD overlaps with parkinsonism, a syndrome of neurological disorders, reducing the diagnostic accuracy of PD to 74–83% [10]. Given the high failure rate of disease-modifying therapies when applied from diagnosis, the field has now refocused its attention to improving early detection and developing biomarkers for tracking progression of PD.

Subtle development of the disease, identified as prodromal PD, has been reported up to 20 years prior to a formal diagnosis of overt motor parkinsonism [11,12,13]. Prodromal symptoms of PD occur in both the central and peripheral nervous system and represent an opportunity for earlier and more accurate diagnosis with the ultimate goal of developing therapeutic interventions [14]. The Movement Disorders Society (MDS) Research Criteria for Prodromal PD define the non-motor symptoms of PD as; REM sleep behaviour disorder (RBD), abnormal results of dopaminergic positron emission tomography (PET), abnormal quantitative motor testing, olfactory loss, constipation, excessive daytime somnolence, symptomatic hypotension, erectile dysfunction, urinary dysfunction, and depression with/out anxiety [14]. A 10-year population-based study was conducted on the performance of the MDS Research Criteria and found that they yielded moderate to high predictive power for incidental PD [15]. A number of individual prodromal symptoms, as well as an additional symptom, mild cognitive impairment (MCI), were also shown to be highly predictive of developing PD [16]. The promising predictive capabilities of these prodromal symptoms have led to the development of many biomarkers and algorithms aiming to identify the early stages of the disease (see reviews [17,18,19]). However, current phenotypic biomarkers are unrefined and have varying levels of sensitivity and accuracy [19]. Another major caveat is that the non-motor symptoms are not specific to a PD diagnosis and currently cannot be used as standalone diagnostic markers. There is an opportunity to explore which high-risk candidate genes lead to prodromal symptoms or non-motor symptoms using animal models. These in-depth genotype-phenotype analyses will be key for establishing which phenotypic markers have utility across species, could identify specific disease-causing biological pathways, and thus contribute to elucidating PD aetiology and the discovery of new therapeutics.

To date, animal models have been integral for understanding the pathological mechanisms of underlying cell loss in the SNpc and subsequent motor impairment in PD. Following significant technological advances, a multitude of genetic and environmental risk factors are now linked to PD and have been introduced into animal models [20,21]. Specifically, high-risk and familial genetic mutations driving disease-causing biological pathways have been explored in animal model systems and these have added insight into how accumulation of misfolded protein aggregates (termed Lewy bodies and neurites), failure of protein clearance, mitochondrial damage, oxidative stress, excitotoxicity, and neuroinflammation all contribute to the disease. These animal models do not aim to replicate every aspect of disease pathology but rather, are designed to elucidate how components of PD pathobiology contribute to the development of motor phenotypes and cell loss. Reflecting a greater emphasis on prodromal PD in the clinic, the assessment of non-motor phenotypes in these animal models has been gaining traction. The degree to which these mouse models present with similar non-motor dysfunction to the clinical condition, termed face validity, requires scrutiny [22]. For these animal models to be useful in mapping risk-factors to biological pathways and disease mechanisms, they must accurately reflect clinical observations and present reliable and reproducible phenotypes.

This systematic review aimed to scrutinise this emerging field by characterising non-motor phenotypes in genetic animal models of PD, including high-risk genes and familial mutations of PD. Animal models utilising toxin-ablation to model cell loss in the SNpc and consequential motor dysfunction were excluded as the focus of this review was to understand the temporal development of a broader range of phenotypes in a whole system akin to clinical PD. Non-motor phenotypes were selected to be equivalent to the MDS criteria, including olfactory loss, constipation, disrupted circadian rhythm (a surrogate for excessive daytime somnolence and sleep disturbances), cardiovascular (hypotension), urinary dysfunction, depression and anxiety, and an additional predictive marker, cognitive impairment. Motor phenotypes and the presence of nigral cell loss were also noted as these represent the accepted standard of PD-like diagnosis in animal models. The specific aims of this review were (1) to identify which phenotypes present most consistently across the animal models, (2) report which phenotypes presented in an age-dependent manner, (3) investigate if animal models recapitulate most non-motor phenotypes and (4) to highlight gaps and provide future recommendations for researchers.

## 2. Methods

This systematic review was conducted in accordance with the Preferred Reporting Items for Systematic Review and Meta-Analyses (PRISMA) [23] and registered with INPLASY (registration code: INPLASY2022110050).

### 2.1. Definitions and Eligibility Criteria

A genetic model of PD was defined as a model that contained a known human genetic mutation or a known genetic risk factor of PD. The non-motor phenotypes were defined as functional assessments of cognition, olfaction, gastrointestinal function, circadian rhythm, cardiovascular function, urinary function, depression, and anxiety, which aligned with the MDS criteria. Studies were included if they (1) used a genetic model of PD, (2) used mice or rats, (3) measured at least one functional outcome from the phenotypes specified, (4) included appropriate controls, (5) contained original work, and (6) were in English. Studies were screened first at the level of title and abstract and exclusions were applied (Figure 1). Following this, full-text articles were examined and deemed eligible or excluded based on the above criteria.

### 2.2. Search Terms and Strategy

Screening and data extraction were performed using the online platform, Covidence (https://www.covidence.org/ (accessed on 14 December 2020)). Pubmed was the primary database utilised and searches were conducted in December 2020 and January 2022. Search terms were: (parkinsonism OR parkinson’s) AND (mouse OR rat OR rodent) AND (olfact * OR hyposmia) OR (circadian rhythm OR RBD OR REM OR sleep) OR (constipation OR gut OR gastrointestinal) OR (anxiety OR depression) OR (cardiovascular) OR (memory) OR (urinary). Once the primary search was complete, and the animal models were identified, a secondary search was performed. This secondary search covered each combination of animal model and phenotype (e.g., A53T AND mouse model AND (olfact * OR hyposmia)). Erectile dysfunction was initially included in the search criteria but resulted in no publications and motor function was not included at this level as the phenotype was extracted once all studies were screened. The total number of references imported for screening (5602) includes references from both primary and secondary searches.

### 2.3. Data Extraction

A template was used for consistent data extraction and are as follows: (1) First author surname, year of publication, title (2) rodent species and sex, (3) Genetic model, background strain, age during experiments (4) main outcome measures methods and results: cell loss, motor, and non-motor behaviour. All studies that met inclusion criteria were also screened for motor assessment and SNpc cell loss to determine if non-motor phenotypes occurred prior to these two late-stage motor and cell loss indicators in a given animal model. Data were sorted into 4-time points over the life of the model (1–5 months, 6–11 months, 7–11 months, and 12+ months) to establish the relative appearance of each phenotype.

### 2.4. Risk of Bias Assessment

Two independent investigators assessed the risk of bias in each study using a modified protocol adapted from the risk of bias tool for animal intervention studies by the Systematic Review Centre for Laboratory animal Experimentation (SYRCLE) [24]. Disagreements were resolved through discussion. Screening bias was not included as this criterion relates to interventions, which are not relevant to this review. Thus, the protocol included: random housing (performance bias), random outcome assessment (detection bias), blinding (detection bias), incomplete outcome data (attrition bias) and selective outcome reporting (reporting bias). The items were categorized into low risk, high risk, unclear risk of bias or NA. NA was given when the bias was not applicable to the study.

## 3. Results

### 3.1. Study Characteristics

In summary, the Movement Disorder Society (MDS) criteria was investigated in 51 different genetic rodent models across 150 studies that satisfied the inclusion criteria (Figure 2A; Table 1; full breakdown in Appendix A). Lesser-known rodent models (*n* = 36) used in 2 studies or fewer were noted, however, not included in the main results due to limited data (results in Appendix B Figure A1; all data in Appendix A). Background strain, specific promotors, sex, and age for each study were not assessed similarly due to limited data (Appendix A). MDS criteria phenotypes were extracted across all studies and the inclusion of motor function and SNpc cell count assessment in each study were noted as these measures have been considered the benchmark progression of neurodegeneration in animal models (Figure 2B). The most investigated MDS criteria phenotype was cognition (79 studies, 52.7% of all included studies), followed by anxiety or depressive-like behaviour (53; 35.3%), olfaction (44; 29.3%), and gastrointestinal (GI) function (23; 15.3%). The least investigated phenotypes were circadian rhythm (15; 10%), cardiovascular (8; 5.3%), and urinary assessments (2; 1.3%). Motor function was concurrently investigated with non-motor phenotypes in 76% of studies (114) however only 18.7% of studies (28) investigated cell loss along with non-motor phenotypes. Just under half of the 150 studies used male rodents (70 studies: 46.7%), 4 studies (2.7%) exclusively used females and 47 used both sexes (31.3%). No sex was specified in 29 (19.3%) studies. Mice were used in the majority of studies (145; 96.7%) and rats were used in 5 (3.3%) (Appendix A). The proportion of studies investigating Movement Disorder Society (MDS) criteria phenotypes for each animal model compared to all studies using the model was calculated to give an indication of how well preclinical research has responded to the shift in defining prodromal PD in the clinic (Figure 2C).

### 3.2. Quality Assessment of Studies

The quality of studies was poor overall (Figure 3). The most unclear risk of bias was in the reporting of random housing (145/150), random outcome assessment (133) and in blinding (90). Only a third of the studies had low risk of attrition bias whilst all but one study had low reporting bias. The highest risk of bias was in reporting of incomplete outcome data (50/150).

### 3.3. How Well Do Genetic Rodent Models of PD Recaptiulate MDS Criteria Phenotypes?

This section aimed to identify if phenotypes presented consistently within animal models, identifying any gaps in specific MDS phenotypes and whether phenotypes presented in an age-dependent manner (For lesser-known models see Appendix A).

#### 3.3.1. Homozygous A53T

The homozygous A53T mouse model was most commonly used to characterise MDS criteria phenotypes, with 40 studies included in this review, representing 15% of all literature using this model (Figure 2C). GI function was assessed in 10/40 studies and dysfunction was highly consistent (10/10) and present at all time points (Figure 4). Cognition was investigated in 16/40 studies, 14 showed differences between mutants and wildtypes and similarly, high prevalence of impairments was seen at all time points (Figure 5). Olfaction was investigated in 10/40 studies, 7 of which reported an impairment and no increase in prevalence was seen with age. Anxiety/depressive-like behaviour was assessed in 12/40 studies, and deficits were seen in only 5 of them and while no age-dependent increase in the prevalence of impairment was observed, several incidents of reduced anxiety/depressive-like behaviour were reported across all age ranges (4/12). Cardiovascular function and circadian rhythm were explored in 2 and 4 studies, respectively and impairments in both were present at 1–5 months. At later ages, circadian rhythm dysfunction was seen in one study (18+ months), however normal cardiovascular function was observed in another at 12–17 months. Overall, 15 of 31 studies assessing motor function in A53T mice observed impairments with increased prevalence across ages. Counter to this, improvements in motor performance were also identified in A53T mice between 1–17 months, although less frequent (6/31). Increasing prevalence of SNpc cell loss was seen from 6 to 18+ months in 4 studies.

#### 3.3.2. Heterozygous A53T

The A53T mutation is a point mutation in α-syn associated with the PD phenotype [175]. Heterozygous/hemizygous A53T were analysed separately from the homozygous model to understand how gene-dosage effects may affect how phenotypes present. Of the 45 A53T studies included in this review, 5 of them used either the heterozygous or hemizygous A53T model, and this accounted for 1.9% of the overall literature using either homozygous or het/hemizygous models (Figure 2). Four studies investigated cognitive function and deficits were only reported by 1 at 6–11 months (1/4) (Figure 4). These results could not be explained by age as a greater number of reports found normal or improved cognitive function at the same time point (Figure 5). No impairments in anxiety/depressive-like behaviour at 6–11 months (0/1), olfaction at 1–5 months (0/1), and circadian rhythm at 12–17 months (0/1) were reported. In contrast, one study explored urinary function between 1 to 17 months and reported reliable impairments across all time points investigated. All 5 studies investigated motor performance, however, just 1 found significant deficits between 6 to 17 months. In contrast, 2 studies found improved motor performance at 6–11 months. One study found significant decreases in SNpc cell counts at 6–11 months.

#### 3.3.3. Homozygous Hα-syn OE

The homozygous Hα-syn OE mouse model which overexpresses wildtype α-syn was assessed for non-motor phenotypes in 22 studies, accounting for 20.2% of the overall Hα-syn OE literature (Figure 2) [176]. Cognition was investigated in 8/22 studies across all time points and impairments were identified in 6 studies, however, were not dependent on age (Figure 4). Anxiety/depressive-like behaviour was explored in 8/22 studies, but only 2 observed impairments at 1–5 months in one and 6–11 months of age in the other (Figure 5). Consistent deficits were found in olfaction (7/7) across all time points, GI function (3/3) and cardiovascular function (1/1) as early as 1mo up to 17 months and circadian rhythm at 6–11 months (1/1). Of the 22 studies investigating the Hα-syn OE model, 15 included motor assessment and 10 of them identified impairments across all time points. The prevalence of deficit was high across all time points except at 6–11 months. Two studies also reported an increased motor phenotype from 1 to 17 months. SNpc cell count was assessed in 4 studies at all time points except 6–11 months and a significant decrease in Hα-syn OE mice was identified only at 12–17 months.

#### 3.3.4. Hemizygous Hα-syn OE (Thy1-αsyn Hemi)

Hemizygous Hα-syn OE were analysed separately from the homozygous line to understand how gene dosage may influence the presentation of phenotypes. Of the 25 Hα-syn OE studies included in this review, 3 of them used the hemizygous model, representing 2.8% of overall Hα-syn OE literature using either hemizygous or homozygous models (Figure 2). Cognition was assessed in 2 studies which showed an age-dependent appearance of deficits from 1 to 17 months (Figure 4). One study identified impairments in both anxiety/depressive-like behaviour and olfaction at 6–11 months (Figure 5). All three studies concurrently assessed motor function and impairments were seen at 1–5 months and 12–17 months, with a further study documenting an improved performance in motor ability at 6–11 months.

#### 3.3.5. A30P

Similar to the A53T model, the A30P is another mutant form of human α-syn used to investigate PD phenotypes [177]. A total of 11 studies examined the MDS criteria phenotypes in A30P mice, which represents 14.7% of all literature examining this mouse (Figure 2). Cognitive function was assessed in over half of these studies across all time points (6/11); however, impairments were only reported at two of these, between 6 to 17 months (2/6) (Figure 4). Consistent impairments were seen in GI function (3/3) and multiple observations were made within these studies demonstrating impairments occurred early and remained throughout aging (Figure 5). Understudied phenotypes include anxiety/depressive-like behaviour (1/1) and circadian rhythm (1/1), and impairments were identified in A30P mice at 1–5 months. Olfactory impairment (1/1) was observed at 6–11 months. Motor function was concurrently assessed in 7/11 studies however 3 of these identified impairments in A30P mice, presenting in an age-dependent manner. Decreased SNpc cell count was identified in one study at 12–17 months.

#### 3.3.6. Mitopark

Mitopark mice is a conditional knockout with disruption of the mitochondrial transcription factor A gene (Tfam) in dopaminergic neurons [178]. MDS criteria phenotypes were investigated in 7 studies using the Mitopark mouse model, representing 18.4% of the wider Mitopark literature (Figure 2). Studies of cognition (3/3), olfaction (2/3), anxiety/depressive-like behaviour (2/2), GI function (1/1) and circadian rhythm (1/2) collectively showed impairments of these phenotypes increasing in prevalence over age (Figure 4). Cognitive deficits appeared earliest at 7–13 weeks, followed by deficits in anxiety/depressive-like behaviour, olfaction and GI function at 14–20 weeks, and circadian rhythm at 21–30 weeks (Figure 5). Of the 7 studies utilising Mitopark mice, 4 of them assessed motor function and found impaired ability (4/4), increasing in prevalence over age and first appearing at 7–13 weeks. SNpc cell count was characterised from 14 to 30 weeks by 2 studies that found significant loss compared to wildtypes.

#### 3.3.7. VMAT2 KO

VMAT2 KO mice express very low levels of the vesicular monoamine transporter 2 protein, a regulator of pre-synaptic dopamine homeostasis [179,180]. Three studies utilising VMAT2 KO mice, representing 25% of all literature, investigated MDS criteria phenotypes (Figure 2). Cognition was assessed by 1 study at multiple time points and impairments were detected from 12–17 months and persisted to 18+ months (Figure 4). Two studies assessed anxiety/depressive-like behaviour and impairments were noted at 1–5 months and 11–17 months but not at 18+ months (Figure 5). Olfaction was investigated by one study across the first 3 time points and found impairments appeared in an age-dependent manner starting at 6–11 months. GI function was assessed in another study at two different time points and found deficits at 1–5 months and 18+ months. No impairment in circadian rhythm was noted at the same time points in this same study, however, another reported improved function in VMAT2 KO mice relative to wildtypes at 1–5 months. Motor performance was assessed in 2 of the 3 studies using VMAT2 KO mice, and 1 identified motor impairment at 12–17 months and 18+ months, whilst the other showed normal performance at 1–5 months and 18+ months.

#### 3.3.8. LRRK2 G2019S

The G2019S mutation is the most common mutation within the LRRK2 gene that is associated with PD [181]. Five studies used the LRRK2 G2019S mouse model to characterise MDS phenotypes, and these accounted for 7.9% of the overall literature (Figure 2). Cognition was assessed in 2 studies and deficits appeared to increase in prevalence from 1 to 17 months (Figure 4). Age-dependent impairments in anxiety/depressive-like behaviour starting from 6–11 months were identified in LRRK2 G2019S mice from 2 studies and circadian rhythm dysfunction was noted at 6–11 months in another study (Figure 5). Whilst deficits in motor performance were reported in only 1 of the 4 studies, the impairments described in this study appeared in an age-dependent manner.

#### 3.3.9. PINK1 KO

Mutations resulting in loss-of-function of the PTEN-induced kinase 1 (PINK1) gene are associated with early onset PD [182]. Eight studies used PINK1 KO rodents, which constituted 42.1% of the literature (Figure 2). Across 3 studies, cognitive function was normal at 1–5 months and impaired at 6–11 months (1/3) (Figure 4). Consistent deficits in anxiety/depressive-like behaviour from 1 to 17 months were reported by 2 studies, whilst 2 other studies reported no differences at 1–5 months and 6–11 months (Figure 5). One study identified olfactory impairments in PINK KO rodents at 18+ months and another showed cardiovascular dysfunction from 1 to 11 months. Motor assessments were made in 6 of the 8 studies and counterintuitively, deficits were present from 1–11 months (2/6) but did not persist at 18+ months. The remaining studies showed no differences in motor performance across the same time points and at 18+ months (4/6). SNpc cell loss was investigated in 1 study at 6–11 months and 18+ months and no significant differences between transgenic and wildtype animals were observed at either time point.

#### 3.3.10. Tau KO

Altered Tau function has been identified as a genetic risk factor of sporadic PD [183]. Most of the literature utilising the Tau KO model included MDS criteria phenotypes in their assessments (66.7%) and minimal impairments were identified (Figure 2). A total of 6/8 studies included cognitive assessments, however, only 1 reported a deficit at 12–17 months (Figure 4). Olfactory dysfunction was identified at 6–11 months and 12–17 months; however, the aged WT group were similarly impaired to the transgenic animals (Figure 5). No differences in circadian rhythm were identified from 1 to 11 months in 1 study. Seven studies also investigated motor ability in Tau KO mice from 6 to 18+ months, and 5 of them described impairments, with the majority of these occurring at 12–17 months. Three studies performed SNpc cell counts, and all found significant loss in Tau KO mice at the same 12–17 months’ time point.

#### 3.3.11. DJ-1 KO

Mutations resulting in loss-of-function of the DJ-1 gene cause early onset PD [184]. DJ-1 KO mice were assessed for MDS criteria phenotypes in 5 studies, representing 9.1% of the overall literature (Figure 2). Cognitive function was assessed in 2 studies from 1 month, and deficits were first reported at the 12–17 months’ time point (1/2) (Figure 4). Impairments in anxiety/depressive-like behaviour were noted from 1 to 11 months of age and no differences were observed from 17 months (2/2) (Figure 5). No impairments in olfaction (0/1) nor cardiovascular function (0/2) were seen at the time points assessed. Of the 5 studies using the DJ-1 KO model, 2 studies investigated motor performance, and both reported deficits between 1 to 11 months (2/2).

#### 3.3.12. LRRK2 R1441G/C

Another mutation of the LRRK2 gene associated with PD is the R1441G/C mutation [185]. Phenotypes in the LRRK2 R1441G or C mutation model were characterised by 4 individual studies, constituting 20% of the literature (Figure 2). No cognitive impairment was reported in the 3 studies which assessed this phenotype from 1–11 months and 18+ months (Figure 4). Anxiety/depressive-like behaviour was assessed in 2/4 studies and impairment was seen in only 1 study at 6–11 months (Figure 5). In contrast, normal behaviour was noted from 6 to 18+ months. Olfaction was also assessed from 6 to 18+ months in 3 studies and deficits were apparent from 12 months (2/3), indicating a contribution of age to the appearance of impairment. Similarly, GI function was assessed from 6 to 18+ months in 1 study and a deficit was found across all investigated time points. Motor function was explored across all time points in 4 studies and significant impairment compared to wildtypes was reported by 3, showing an increased prevalence of impairment over age. One study performed SNpc cell counts but no significant differences were seen between the two genotypes at 1–5 months nor 18+ months.

#### 3.3.13. CD157 KO

CD157/BST1 is a risk locus for PD and the CDK157 KO is model of the psychiatric phenotypes of PD [160]. There were 4 studies that examined MDS criteria phenotypes in the CD157 KO mice, and this represented 100% of the CD157KO/BST1 rodent literature (Figure 2). This model predominantly presents with anxiety-like impairments, and thus, this was the main phenotype investigated. All 4 studies determined anxiety/depressive-like behavioural deficits in this model at 1–5 months (Figure 4). Improved cognitive function, as well as dysfunctional circadian rhythm, was seen at 1–5 months (Figure 5). Motor function was also assessed at 1–5 months in 2 studies and deficits were reported in only one.

#### 3.3.14. LRRK2 KO

The LRRK2 KO mouse model also recapitulates the loss of function due to mutations within the LRRK2 gene [186]. Non-motor phenotypes in the LRRK2 KO model were examined in 5 studies, accounting for 45.5% of the broader LRRK2 KO literature (Figure 2). Normal cognitive function was seen from 1 to 17 months across all 3 studies assessing the phenotype (Figure 4). In contrast, deficits were found in anxiety/depressive-like behaviour from 6 to 17 months by 1 study (Figure 5). Olfaction was investigated at 1–5 months and 18+ months in 2 studies, and results showed impairments were only present at 1–5 months (1/2). One study investigated GI function at 1–5 months and did not find deficits. Out of the 5 studies investigating LRRK2 KO mice, 3 concurrently assessed motor function across all time points, and none reported any significant differences between LRRK2 KO and wildtype animals.

#### 3.3.15. Parkin KO

Mutant Parkin genes cause autosomal recessive PD [187]. Seven studies investigated MDS criteria phenotypes in the Parkin KO model, constituting 50% of the overall literature (Figure 2). Cognition was explored by 6/7 studies across all time points, and 4 studies determined deficits were present from 1 to 17 months (Figure 4). Anxiety/depressive-like behaviour was evaluated in 3 studies across all time points and impairments were identified in 2 studies from 6 to 18+ months (Figure 5). Deficits were neither found in olfaction from 1 to 11 months (0/1) nor in cardiovascular function at 1–5 months (0/1). Motor function was explored in 5 out of the 7 studies and impairments were identified in 2 studies across all time points. Lastly, 1 study showed no difference in SNpc cell counts between the genotypes at 12–17 mo months.

### 3.4. Which Phenotype Is Most Consistent across All Animal Models?

The consistency of phenotypic outcomes across multiple animal models was assessed independent of age (Figure 6). GI deficits were highly consistent across animal models (95% of studies found deficits), followed by olfaction (70.5%) and circadian rhythm (61.5%). Further, 57.8% of studies found deficits in cognition and 55.6% of studies reported deficits in anxiety/depressive-like behaviour. Less than half of the studies observed cardiovascular dysfunction in their rodent model (42.8%). Only one study investigated and found deficits in micturition reflexes (urinary function). Deficits in motor function and reductions in SNpc cell numbers were also reported in 51–55% of studies.

## 4. Discussion

The current review addressed a critical gap in the literature by assessing the prevalence and consistency of non-motor phenotypes (cognition, olfaction, GI function, anxiety/depressive-like behaviour, circadian rhythm, cardiovascular and urinary function) in genetic PD rodent models. Phenotypes were scrutinised for consistency across all rodent models, and GI function (95% of studies showed deficit) and olfaction (70.5%) were the most well recapitulated. This finding is relatively consistent with the clinical literature, as the prevalence of olfaction is up to 90% and GI function approximately 65% of people with PD [188,189]. The degree to which the animal models closely replicated all phenotypes relevant to PD was inconclusive due to poor reproducibility. Understudied phenotypes, animal models, and ages represented the greatest gaps in the literature.

### 4.1. The Contribution of Variability to the Reproducibility of Phenotypes

Reproducibility in results is the ability to repeat a study independently and draw similar conclusions and is influenced by a number of environmental and experimental variabilities across labs, animal cohorts, and in methodological assessments [190,191]. Many phenotypes across genetic PD models had low reproducibility, with the exception of GI dysfunction and olfactory deficits. Cognitive impairments and anxiety/depressive-like behaviour were the least reproducible and this might represent greater susceptibility to environmental and experimental variability [192,193].

#### 4.1.1. GI Function

GI dysfunction is apparent in approximately 65% of people with PD, with α-syn pathology manifesting across the enteric nervous system which governs GI function within the prodromal phase [194]. In this review, GI dysfunction was identified as the most consistent phenotype, appearing in almost all PD rodent models where it was assessed. It was common for studies to use more than one measure of GI function, which increased within-laboratory reliability and degree of certainty regarding the outcome. These outcome measures included the number of faecal pellets, stool weight and in vivo and in vitro measures of gut motility/transit time. An example of this can be seen in one study using A53T mice where opposite directions of effect were found in their two measures of GI function across two separate cohorts. In one cohort, the authors show a reduced number of contractions in in vitro colonic motility recordings however in the other, they reported increased in vivo faecal pellet output [50]. The authors physically restrained mice for oral gavage prior to the in vivo test, a known stressor that has been shown to increase colonic motility [195,196]. Another study assessing A53T mice on a longitudinal single housing stress paradigm showed a similar disconnect using two in vivo measures, reporting increased bead expulsion time but no change in faecal pellet output compared to A53T in group-housed conditions [35]. While this review did not scrutinise the consistency of multiple outcome measures used in the same study, these examples of the nuanced effects of stress highlight the importance of experimental design to mitigate variance. Overall, the consistency of impaired gut function across all genetic PD mouse models, regardless of assessment method may reflect a common disrupted pathway and thus represents an opportunity for PD-specific treatments for GI dysfunction.

#### 4.1.2. Olfaction

The clinical prevalence of hyposmia is as high as 90% and can manifest 20 years prior to diagnosis [197]. In the current review, olfactory dysfunction mimicking clinical observations was identified in just under half of PD mouse models, with 4 of them showing inconsistent findings. The A53T and Hα-syn models both showed consistent olfactory deficits prior to motor impairments and could represent useful tools for the further investigation of PD-specific mechanisms driving olfactory dysfunction. An important consideration in evaluating the reproducibility of hyposmia in mouse models is the use of specific olfactory cue types in assessment methods that activate the two distinct rodent olfactory systems. Non-social cues, for example, food or essential oil fragrances, activate the main olfactory bulb system, the analogous structure to the human olfactory system. Social cues including pheromones of opposite-sex urine, predator scents, or used bedding target the vomeronasal system which is responsible for the detection of pheromones, governing sexual and mating behaviour [198]. In humans, α-syn aggregates and atrophy have been observed in the olfactory system [199,200,201] and thus, impairments in rodent olfactory tests utilising non-social cues are considered more analogous to olfactory deficits in PD. Of the studies investigating olfaction in this systematic review, 75% of those using non-social cues reported deficits. Interestingly, 58% of studies using social cues and thus assessing vomeronasal function, also found deficits. As the human vomeronasal system is largely reduced and has not been investigated in a PD setting, the translational relevance of impairments in this mouse-specific system can be questioned [202]. Given these unknowns, using social cues exclusively may introduce ambiguity and the use of non-social and social cues for comprehensive phenotyping, or exclusively non-social cues is recommended, as this targets the known analogous olfactory structures in PD.

#### 4.1.3. Cognition

PD with mild cognitive impairment (PD-MCI) and PD with dementia (PDD) are defined by deficits in attention, executive function, visuo-spatial function, (long-term/recall) memory, and language [203]. In the current systematic review, cognition was the most assessed MDS criteria phenotype (79 studies), however, not all cognitive domains were assessed with visuospatial memory tasks overrepresented. Significant gaps in understanding attention and executive function exist, with only one study using a reversal learning operant chamber paradigm to assess executive function in the Hα-syn OE (Thy1-αsyn) mice [83]. Differences in mouse vocalisations have been used by the field as a proxy for language dysfunction in humans, and these were shown to be impaired in one study also using Hα-syn OE (Thy1-αsyn) mice [87]. However, as vocalisations are yet to be validated as a sophisticated cognitive function, the utility of these vocalisations remains an area for further expansion [204]. Comprehensive cognitive phenotyping to address the significant gap in the characterisation of PD cognitive impairments in animal models is required. Current clinical descriptions of PD point to specific patterns of cognitive impairments and replicating clinical ‘subcortical’, defined by greater deficits in executive, visuo-constructive, and attention, or ‘cortical’ cognitive types with more severe memory impairments in animal models may shed light on potentially distinct pathological pathways [205].

While cognitive dysfunction in PD rodent models receives significant attention in the field, consistent impairments are not reported. This is in part due to the limited sensitivity of the cognitive tasks used and experimental and environmental variabilities [206]. Methodological and apparatus design differences within the same test can alter effect sizes and behavioural results, and in some cases may even assess different cognitive domains [207]. For example, the novel object recognition task is the most commonly used cognitive task in PD animal models, and variable inter-trial intervals (ITI) ranging from 5 min to 24 h, target working memory, and long-term reference memory, respectively. These impairments are frequently reported using the umbrella term of “cognitive dysfunction” leading to difficulty in resolving the PD-specific cognitive ability assessed (e.g., working memory or spatial memory). More explicit reporting of specific cognitive capability being assessed and careful consideration of whether the intended protocol is measuring the correct cognitive domain is imperative to improving the discrepancies within animal models and thus the translation to clinical outcomes. As is the nature of cognitive behavioural tasks, certain cognitive tests are inherently more sensitive to external factors, leading to differential outcomes despite the same parameters. A meta-analysis on the Morris water maze (MWM) test revealed differences in genetic background strains and environmental factors were enough to create variation in behavioural outputs that could mask a true effect of a mutation [192]. Further, a multi-laboratory study comparing two mouse background strains, C57BL/6NCrl and DBA/2NCrl, identified the overwhelming contributor to cognitive differences in these strains were environmental effects and laboratory factors [208,209]. Optimising the protocol for the specific background strain and laboratory environment may represent a way to standardise across different animal models and to improve reproducibility of results [210]. Another consideration when designing cognitive tests for PD animal models is that tasks reliant on motor function for completion coupled with time-based outcome measures, such as the MWM, are not appropriate for PD models. This introduces another source of variability as results are confounded by the differing onset of motor impairments. While concurrent motor tests can aid in clear interpretation of cognitive deficits in absence of motor confounds, these tests do not have the sensitivity to identify subtle motor changes. Instead, tests measuring performance indicators such as the percentage of trials correct, such as operant and touchscreen testing may be more suitable, especially for PD animal models showing significant motor impairment. Specifically, touchscreen testing has emerged as a sensitive and translatable tool enabling the measurement of multiple cognitive domains of relevance to human cognition within the same testing apparatus [211].

#### 4.1.4. Anxiety/Depressive-Like Behaviour

Anxiety and depression present up to 20 years prior to diagnosable motor symptoms and people with PD have a significantly greater frequency of developing psychiatric conditions than the general population [212]. Many tests exist to assay these conditions in PD animal models, including Elevated Plus Maze (EPM), Light-Dark Box (LDB) and Open Field for anxiety-like behaviour, and Forced Swim Test (FST), Tail Suspension Test (TST) and Sucrose Preference Test (SPT) for depressive-like behaviour. In the present review, it was not possible to conclusively determine if the majority of PD animal models recapitulated PD anxiety or depressive-like phenotypes as over half of the animal models assessed with these testing paradigms showed highly inconsistent phenotypic outcomes or no-change relative to controls. Assessment of anxiety and depressive-like behaviour in rodents represents a significant challenge as a multitude of factors have been shown to modulate their expression. These include sex of the experimenter, amount and type of handling methods, lighting, long-distance transportation and inadequate habituation and room smells [191,213,214,215,216,217]. A concrete example of this is that increased stress can induce hyperlocomotion, resulting in increased swimming time in the FST, and the interpretation of ‘decreased depressive-like behaviour’ [218,219]. The validity of these assays in accurately measuring anxiety/depressive-like behaviour has been questioned, with suggestions that performance could reflect fear-induced avoidance (EPM) or behavioural adaptations to survival (FST) instead of the phenotypes intended [220,221,222]. Standardising methodological assessment and minimising stressor confounds has shown to reduce inter-subject variability and boost the sensitivity of these assays [193,223,224,225]. However, it is possible that anxiety or depressive-like behaviour cannot be modelled in rodents as many of these one-off tests are simply not robust enough to accurately replicate a clinical PD symptom which is also strongly modulated by the environment. It would be powerful to combine these modes of assessment with in vivo physiological measurements of corticosterone, challenges to the hypothalamic-pituitary axis, or assay response to clinically effective treatments like SSRIs.

#### 4.1.5. Understudied Phenotypes

The present review revealed significant gaps in research focus on excessive daytime somnolence (EDS) (circadian rhythm), symptomatic hypotension (cardiovascular function) and urinary function in PD animal models. These symptoms have been documented widely in the PD population and their presence alongside other prodromal symptoms increases the probability of prodromal PD [14]. Thermoregulatory and urinary dysfunction were correlated with CSF biomarkers and patients without evident dopaminergic dysfunction presented with more severe autonomic dysfunction than people clinically diagnosed with PD, indicating dysfunction occurring prior to overt motor impairments [226]. Given their utility as PD-specific biomarkers, these autonomic phenotypes should not be overlooked in preclinical research. Circadian rhythm dysfunction was inconsistently reported across 9 PD models, cardiovascular function was impaired early in 3 PD mice but not another 2, and urinary dysfunction was reported in 2 PD models. Further investigations in mouse models to identify these autonomic phenotypes are needed to ascertain how genetic risk factors may result in EDS, cardiovascular and urinary dysfunction.

### 4.2. Tracking Age-Dependent Phenotypes to Understand Different Pathological Trajectories

In human PD, non-motor symptoms can precede the hallmark motor impairments at diagnosis by 20 years and provide an opportunity to implement predictive biomarkers and early therapeutics. However, a current challenge in the development of phenotypic biomarkers is the non-specificity of these symptoms to PD. For animal models to recapitulate these non-motor symptoms, establishing the temporal occurrence of prodromal symptoms prior to motor dysfunction is essential. Animal models that achieve this will be useful to uncover biological biomarkers, shed light on early disease mechanisms and raise the profile of these non-motor symptoms as predictive clinically observable markers of PD.

In this review, two animal models, A53T and Mitopark mice showed potential and surprisingly distinct patterns of age-dependent phenotypes. Deficits in cognition, GI function, circadian and cardiovascular function were apparent in A53T mice before 6 months of age, whilst nigral cell loss was significant after 6 months. However, motor dysfunction was inconsistently reported until 18 months of age and thus did not appear to be a good indicator of apparent nigral degeneration. The appearance of early stage non-motor phenotypes is in concordance with various early stage pathology found within the hippocampus, colon and ENS, and retinal ganglion cells in these mice [44,47,55,227]. Conversely, in Mitopark mice, cognitive and motor deficits appeared first and impairments in anxiety/depressive-like behaviour, olfaction, and GI dysfunction presented afterward. Circadian rhythm dysfunction only appeared in the advanced stage of pathology; an interesting finding given circadian rhythm dysfunction is a widely documented early predictive biomarker of PD [16]. Further, Mitopark mice had consistent appearance of motor impairments in an age-dependent manner compared to A53T. Cautious interpretation of these results is warranted given the limited data across the ages. Nevertheless, this preliminary finding poses an interesting avenue to explore differences in phenotypic expression between the two genetic risk factors as this may be reflective of prodromal subtypes seen in people with PD [228]. Moving forward, the reliability of phenotype comparisons across age in animal models may be improved by conducting all behavioural assays within the same cohort, although care should be taken to select assays that avoid test re-test effects and confounds of stress due to improper testing order [229]. Longitudinal assessment of non-motor phenotypes for many animal models is a significant literature gap and is recommended as a focus for future studies.

### 4.3. Do PD Rodent Models Have Good Face Validity?

While historically a good animal model of PD should demonstrate motor impairment and nigrostriatal degeneration, with the recent inclusion of prodromal symptoms in the research diagnostic criteria of PD, non-motor phenotypes should also be included [14]. While it is unreasonable for the face validity of a model to recapitulate PD in its entirety, to date, assessments of non-motor symptoms have been overlooked. The inclusion of non-motor phenotypes provides opportunities for researchers to target therapeutics for specific non-motor phenotypes as these significantly decrease quality of life for people with Parkinson’s [230]. The most studied mouse models, A53T and Hα-syn OE mice were consistent in reproducing most of the assessed non-motor phenotypes which present an opportunity to understand how these genetic risk factors lead to widespread PD-like symptomology. However, significant inconsistencies in reproducing motor impairments and nigral cell loss were identified between laboratories and cohorts in A53T and Hα-syn OE mice, although these assessments were not conducted alongside other behavioural testing in all studies. The low reproducibility of the hallmark motor and cell loss phenotypes of PD across laboratories reinforces the requirement for sensitive, reliable, and reproducible assays that are robust against environmental variation. Given the environmental variation between laboratories and cohorts is inevitable, concurrent assessment of nigral cells, and motor and non-motor phenotypes may further improve the benchmarking of neurodegeneration in these models and aid in comparisons between studies. The overall face validity of the other animal models could not be determined due to insufficient data across the non-motor phenotypes. However, preliminary evidence suggests that the models presented with varying numbers of different phenotypic dysfunction. No one model can replicate the entirety of PD, thus, models with consistent deficits in a small number of non-motor phenotypes as well as motor and nigral cell loss constitute useful tools to investigate specific pathway mechanisms.

The variable phenotypic profiles of PD animal models could reflect the heterogeneity of PD and represent good face validity. Clinical observations and cluster analyses have defined 4 PD subtypes based on patterns of motor and non-motor symptom clusters and genetic mutation subtypes are symptomatically differentiated from one another [231,232,233]. Could early onset, rapid disease progression, tremor dominant, and non-tremor dominant subtypes of PD be linked to specific risk variants in PD animal models [234,235,236]? People with A53T mutations develop an early onset aggressive form associated with cognitive impairment whilst those with LRRK2 mutations had less severe deficits in cognition and olfaction compared to idiopathic PD [237,238]. Interestingly, these symptoms appear to align with rodent models containing these mutations. Nevertheless, the symptom and pathological variance support the need for a diversity of animal models to uncover how different aetiologies develop into PD. Further, given the dominant environment and gene interaction hypothesis of PD, introducing an environmental exposure to a genetic risk factor may align rodent models closer to a PD subtype [239]. The heterogeneity in PD may not be a limitation in translating from animals to humans, but rather, represent tools to stratify subtypes of PD using specific risk variants and sophisticated phenotyping approaches. Determining good face validity in animal models with a subset of non-motor phenotypes would initiate opportunities for individualistic treatment of PD [240]. Current failures in treatment could stem from the classification of PD as a single entity and precision medicine targeting individual subtypes is a promising solution. Investigating underlying mechanisms using genetic risk models is an important first step to optimising precision medicine and may expand personalised treatments to broader idiopathic forms of the disease [233].

### 4.4. Limitations

In the current review, different background strains, promotors, sex, and testing paradigms were collapsed due to limited data, prohibiting interpretation of how these influence expression of phenotypes in animal models [190,192]. This review was restricted to motor and cell counts from the studies also assaying non-motor phenotypes and is therefore limited in commenting on the reliability and consistency of these phenotypes across the entirety of this literature. Given the emerging nature of this field, a meta-analysis was not achievable due to low numbers of reports for some phenotypes (e.g., RBD/REM sleep behaviour disorder, cardiovascular hypotension, or urinary and erectile dysfunction) and for animal models across ages.

### 4.5. Recommendations and Opportunities

We systematically investigated all rodent models harbouring PD-associated genetic risk factors and evaluated the degree to which they recapitulated MDS-criteria phenotypes. Highly consistent MDS-criteria non-motor phenotypes including GI dysfunction were identified across all models, representing an opportunity to understand common pathological pathways. Gaps in our understanding in a number of areas were uncovered and similarly represent an opportunity for further research. This review also uncovered a limited ability to reproduce phenotypes within animal models and we comment on approaches to improve rigor in behavioural neuroscience methodology and experimental design (Table 2).

## 5. Conclusions

Animal model systems are used to advance mechanistic understanding of PD and trial experimental therapies, however, the current status quo in measuring motor and cell loss outcomes will not target these investigations to earlier stages of the disease. We must see a reliable recapitulation of MDS criteria phenotypes in these animal systems to reflect current clinical observations. This review identified highly consistent MDS-criteria non-motor phenotypes to target for early stage research, specifically GI dysfunction. Significant gaps for further exploratory study include the understudied phenotypes, circadian rhythm, cardiovascular and urinary dysfunction, and an understudied number of animal models and age ranges. The unique phenotypic profiles of rodent models may reflect the heterogeneity in PD and thus might model different PD subtypes. These studies, reflecting diverse genetic risk factors could be useful in uncovering distinct therapeutic targets, potentially leading to personalised treatments for people with PD. Increased rigor in behavioural neuroscience methodology and experimental design are required to improve reproducibility of phenotypes between different laboratories. The adoption of new methods of assessment with clinical relevance is emerging as an approach to capture previously difficult to-measure phenotypes like executive dysfunction. A shift in the focus of PD preclinical animal models to include a wider range of phenotypic measures that more accurately reflect the clinical description of PD will ultimately improve the back-translation of findings and produce reliable tools for not only identifying new targets for treatments but screening the efficacy of these.

## Figures and Tables

**Figure 1 biomedicines-10-03026-f001:**
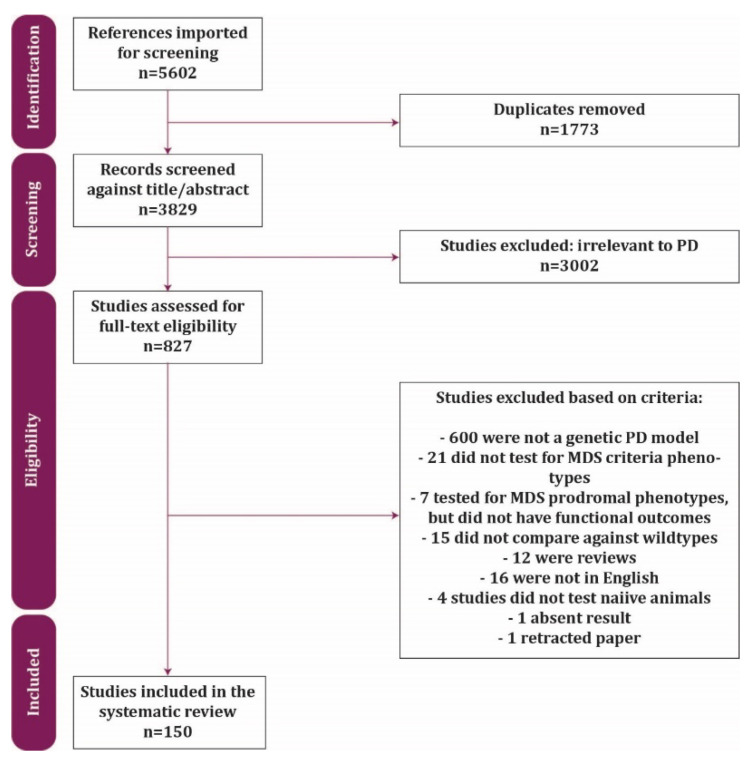
Flow chart following PRISMA guidelines of the selection process and reasons for exclusions.

**Figure 2 biomedicines-10-03026-f002:**
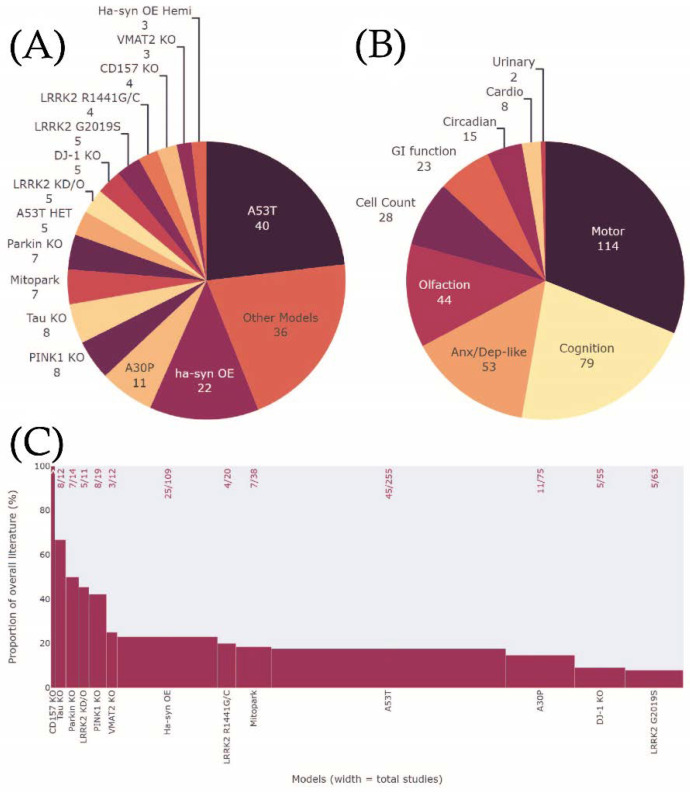
Pie chart of (**A**) the range and number of animal models identified via Pubmed search. (**B**) The number of phenotypes investigated by the studies. (**C**) Proportion of studies investigating MDS criteria phenotypes compared to all published studies using the animal model. The width of *x*-axis represents the total number of studies fitting the inclusion criteria, and *y*-axis represents proportion of overall literature using the model. ^CD157KO proportion = 4/4.

**Figure 3 biomedicines-10-03026-f003:**
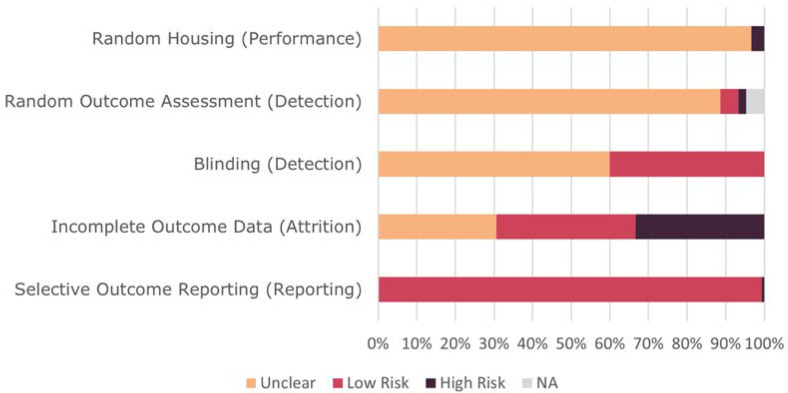
Risk of bias assessment summary. NA results of random outcome assessment refer to studies which tested mice simultaneously.

**Figure 4 biomedicines-10-03026-f004:**
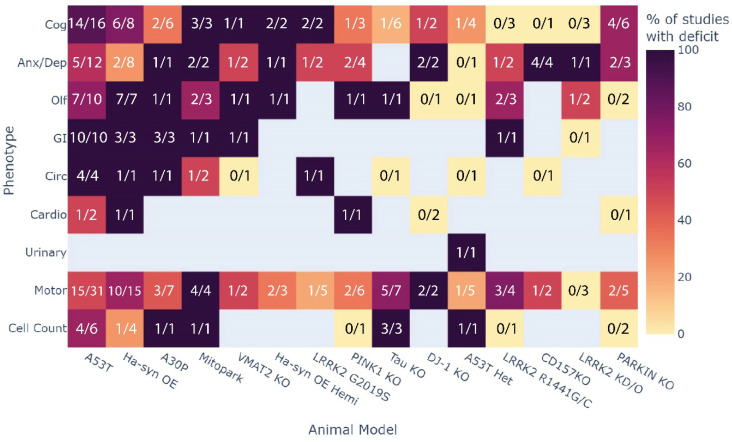
Heatmap summary of the presentation of all phenotypes across all animal models with at least 3 studies. Colours represent the percentage of studies that showed an expected impairment of the phenotype where dark purple is 100% and pale yellow is 0% of studies. The fractions indicate the number of studies with impairment out of all the studies that investigated the phenotype., e.g., 6 out of 8 studies showed impaired cognition in the Hα-syn OE model.

**Figure 5 biomedicines-10-03026-f005:**
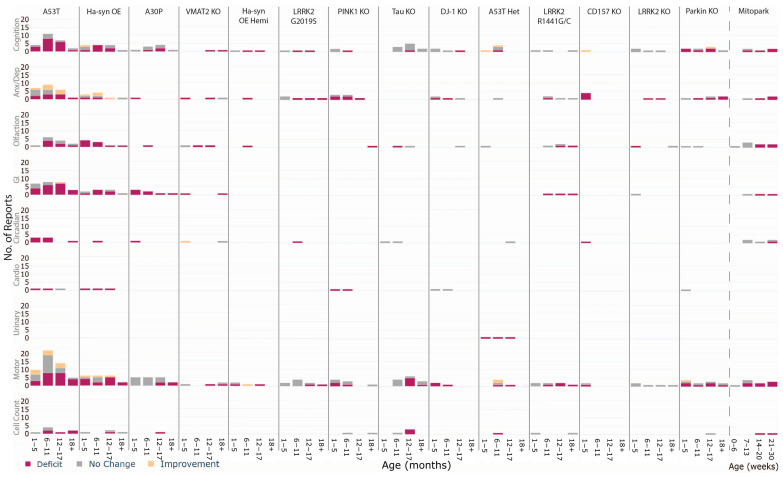
Age-dependent impairments in PD rodent models. Stacked bar chart of phenotypic observations showing deficits (aligning with human PD), no change, and improvement (opposite effect to expected). Number of observations are binned across age brackets (1–5; 6–11; 12–17 and 18+ months of age or 0–6; 7–13; 14–20; 21+ weeks) and observations can represent many from a single study.

**Figure 6 biomedicines-10-03026-f006:**
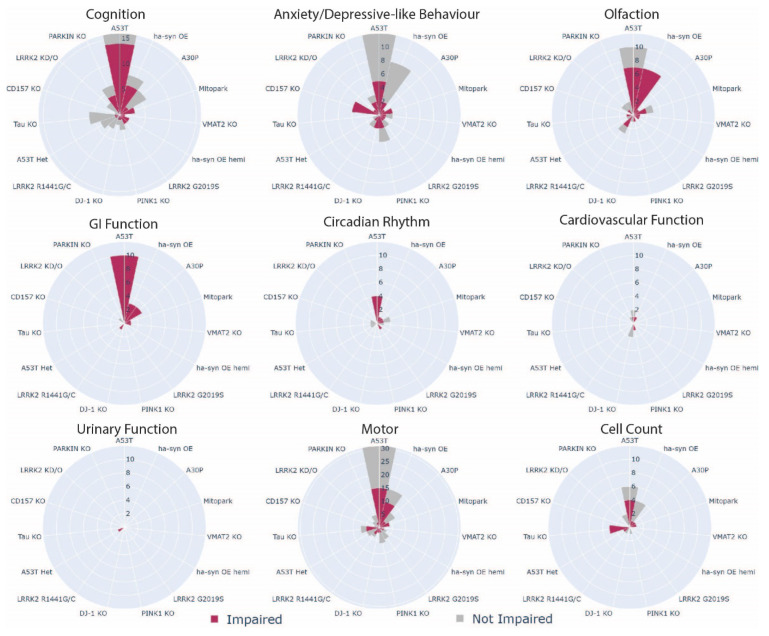
Wind rose charts of the incidence of MDS criteria, motor, and cell count phenotypes across PD rodent models. Radial axis is the number of studies. Numbers in red is a significant impairment of the phenotype, grey is no change or an improvement of the phenotype. The overall wedge height of the bar (red + grey) is the total number of studies for the animal model. Note max radial axis of ‘Motor’ is 31, ‘Cognition’ is 16, and the remaining are set at 12.

**Table 1 biomedicines-10-03026-t001:** All genetic models and references included in the review. Results from lesser-known models used in 2 or fewer studies were not included in the main results and can be found in Appendix A.

Models in Main Results	Ref. No.	Lesser-Known Models	Ref. No.	Lesser-Known Models Cont.	Ref. No.
Homozygous A53T	[25,26,27,28,29,30,31,32,33,34,35,36,37,38,39,40,41,42,43,44,45,46,47,48,49,50,51,52,53,54,55,56,57,58,59,60,61,62,63,64]	A30P/A53T	[65,66]	Adh4 KO	[67]
Homozygous human alpha-synuclein (hα-syn OE)	[45,49,68,69,70,71,72,73,74,75,76,77,78,79,80,81,82,83,84,85,86,87]	PINK KO/A53T	[28]	LRRK2 OE	[88,89]
A30P	[26,29,46,90,91,92,93,94,95,96,97]	GBA+/−/A53T and GBA+/−	[39]	GPR37 KO	[98,99]
Mitopark	[100,101,102,103,104,105,106]	Tau KO/A53T	[43,46]	DAT:TH KO and DAT-DTR	[107]
VMAT2 KO	[108,109,110]	α-syn/GBA+/−	[111]	GDNF-deficient	[112]
hα-syn OE Hemi	[113,114,115]	αβγ-syn KO	[116]	MDK KO	[117]
LRRK2 G2019S	[89,118,119,120,121]	α-syn n103	[122]	VMAT2 Het	[123]
PINK1 KO	[28,124,125,126,127,128,129,130]	hα-syn TP and hα-syn 119	[131]	En1+/−	[132]
Tau KO	[43,46,133,134,135,136,137]	SNCAS129A and SNCAS129D	[138]	B4gInt1 KO	[139,140]
DJ-1 KO	[141,142,143,144,145]	Park KO/TauVLW and TauVLW	[146]	c-rel KO	[147]
A53T Het	[148,149,150,151,152]	LRRK2 R1441G/TauP301S and TauP301S	[153]	Cul9/Parkin KO and Cul9 KO	[154]
LRRK2 R1441G/C	[153,155,156,157]	TauP301L	[158]	SEPT4+/−	[159]
CD157 KO	[160,161,162,163]	Tau+/−	[137]	Id2 KO	[164]
LRRK2 KD/O	[88,89,155,165,166]	TauV337M hemi	[167,168]		
Parkin KO	[146,154,169,170,171,172,173]	Adh1 KO and Adh1/4 KO	[174]		

**Table 2 biomedicines-10-03026-t002:** Summary highlight of recommendations and opportunities.

**Opportunity to fill knowledge gaps in the areas of circadian, cardiovascular, and urinary phenotypes**
These were highlighted as the least assessed MDS criteria phenotypes across all models.
**Characterising the age-dependent appearance of phenotypes within animal models enables understanding of how and when genetic risk factors affect the whole system**
The A53T and Mitopark models suggest differential trajectories of pathology. Further research into other models over age may unveil different subtypes that may align with clinical subtypes.
**Investigating common mechanisms underlying gastrointestinal dysfunction**
Highly consistent GI dysfunction across multiple models represents an exciting target to investigate and is also highly prevalent in clinical PD.
**Consider the construct validity of phenotypic tasks**
For example, using either non-social cues or both social and non-social cues in olfactory tests is better suited to targeting the main olfactory system which is clinically relevant to human PD.
**Consider the methodological translatability of assessments to clinical PD**
Clinical literature suggests heterogenous cognitive profiles in people with PD which represents an opportunity to extend cognitive assessments in PD mouse models to executive function, attention and language, and link underlying neuropathology to the specific cognitive domains.
**Improving rigor in experimental design to reduce the effect of environmental variabilities**
Variability across laboratories is significant and reduces the reproducibility of phenotypes, especially in behavioural tests susceptible to the environment. Within study and between study variations have been shown to have little effect on the phenotypic reproducibility, therefore, reducing stressor confounds, optimising protocols within individual cohorts, and performing thorough characterisation of multiple phenotypes may represent some solutions to improving inconsistent results.

## Data Availability

The data presented in this study are available within the article and Appendix A.

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
