# Peer review of "How Well Do Rodent Models of Parkinson’s Disease Recapitulate Early Non-Motor Phenotypes? A Systematic Review"

_biomedicines, 2022, doi:10.3390/biomedicines10123026_

Round 1

Reviewer 1 Report

Review of a manuscript :”How well do rodent models of Parkinson’s disease recapitulate early non-motor phenotypes? A Systematic Review” by Tracy D Zhang and coauthors submitted to “Biomedicines”

Parkinson’s disease is a second after Alzheimer’s disease neurodegenerative disorder affecting about six million people in the world. Currently there is neither treatment changing the main course of the disease nor reliable biomarkers for the early diagnostic of the disorder. In the manuscript the authors performed a systematic literature search and summarize published data about existing and future opportunities to improve phenotype reproducibility with an emphasis on phenotypic assay choice and robust experimental design. This is a thought-provoking review which will be interesting for the readers of “Biomedicines”.

The following corrections should be done.

Abstract

Line 24. “The A53T and hα-syn overexpression (OE) models recapitulated the majority of phenotypes…”

This is inaccurate sentence, because A53T mutants are also human. The authors should clarify what they mean. Probably, “The human wild-type and A53T mutant α-syn overexpression (OE) models recapitulated the majority of phenotypes…” ?

Introduction

Line 42:”Current diagnosis relies on the development of hallmark motor symptoms of bradykinesia, rigidity, and tremors [6].” The authors should add here a reference on a new review:”Biomarkers in Parkinson’s Disease”. Chapter in a book Peplow P.V., Martinez B., Gennarelli T.A. (eds). Neurodegenerative Diseases Biomarkers. 2022. Neuromethods, vol 173. pp 155-180. Humana, New York, NY. https://link.springer.com/protocol/10.1007/978-1-0716-1712-0_7

Lines 57-58: ”The Movement Disorders Society (MDS) Research Criteria for Prodromal PD define the non-motor symptoms of PD as; REM sleep behavior disorder (RBD), abnormal dopaminergic positron emission tomography (PET)…”This is an awkward sentence. Dopaminergic positron emission tomography (PET) cannot be a symptom of PD. The sentence should be corrected as follows : ”The Movement Disorders Society (MDS) Research Criteria for Prodromal PD define the non-motor symptoms of PD as: REM sleep behavior disorder (RBD), abnormal results of dopaminergic positron emission tomography (PET)…”

Results:

Line 152: ”In summary, 51 different genetic rodent models across 150 studies satisfied the inclusion criteria and investigated the Movement Disorder Society (MDS) criteria…” The sense of the sentence is unclear. Probably, the authors mean :" In summary, 51 different genetic rodent models across 150 studies satisfied the inclusion criteria and corresponding the Movement Disorder Society (MDS) criteria…??

Lines 220-221: ”3.2.2. Heterozygous A53T”

“Heterozygous/hemizygous A53T were analysed separately from the homozygous model to understand how gene-dosage effects may affect how phenotypes present.” The authors should give the following definition of A53T for those readers who may be unfamiliar with the term: ”A53T Mutation is a point mutation in a-synuclein associated with PD phenotype.

Lines 415. “In this review, GI dysfunction was identified as the most consistent phenotype across almost all PD rodent models assessed.” The authors should explain, at least hypothetically why GI dysfunction is the most consistent?

Author Response

Abstract 

Line 24. “The A53T and hα-syn overexpression (OE) models recapitulated the majority of phenotypes…” This is inaccurate sentence, because A53T mutants are also human. 

We have corrected this sentence to read “The mouse model harbouring mutant A53T, and the wild-type hα-syn overexpression (OE) model recapitulated the majority of phenotypes". 

Introduction 

Line 42:”Current diagnosis relies on the development of hallmark motor symptoms of bradykinesia, rigidity, and tremors [6].” The authors should add here a reference on a new review. 

We have added this reference. 

Lines 57-58: ”The Movement Disorders Society (MDS) Research Criteria for Prodromal PD define the non-motor symptoms of PD as; REM sleep behavior disorder (RBD), abnormal dopaminergic positron emission tomography (PET)…” This is an awkward sentence. 

We have corrected this sentence to the suggested phrasing: “The Movement Disorders Society (MDS) Research Criteria for Prodromal PD define the non-motor symptoms of PD as: REM sleep behaviour disorder (RBD), abnormal results of dopaminergic positron emission tomography (PET)…"  

Results: 

Line 152: ”In summary, 51 different genetic rodent models across 150 studies satisfied the inclusion criteria and investigated the Movement Disorder Society (MDS) criteria…” The sense of the sentence is unclear. 

We have clarified this sentence to read “In summary, the Movement Disorder Society (MDS) criteria was investigated in 51 different genetic rodent models across 150 studies that satisfied the inclusion criteria.” 

Lines 220-221: ”3.2.2. Heterozygous A53T” 

“Heterozygous/hemizygous A53T were analysed separately from the homozygous model to understand how gene-dosage effects may affect how phenotypes present.” The authors should give the following definition of A53T for those readers who may be unfamiliar with the term. 

We have added the definition as "The A53T mutation is a point mutation in α-syn associated with the PD phenotype."

Lines 415. “In this review, GI dysfunction was identified as the most consistent phenotype across almost all PD rodent models assessed.” The authors should explain, at least hypothetically why GI dysfunction is the most consistent? 

We have added further clarification to the sentence to read “In this review, GI dysfunction was identified as the most consistent phenotype, appearing in almost all PD rodent models where it was assessed.”  

Reviewer 2 Report

This is a very detailed study concerning the non-motor symptoms in rodent models of Parkinson’s disease.

Some minor comments:

Please, define and describe shortly the investigated models for the sake of general readers. Alpha-synuclein overexpressing or A53T mutant alpha-synuclein models are easy to understand, but what kind of model is Mitopark or CD157?

Are Tau models specific for Parkinson’s disease or for Alzheimer’s as well?

Author Response

Please, define and describe shortly the investigated models for the sake of general readers. Alpha-synuclein overexpressing or A53T mutant alpha-synuclein models are easy to understand, but what kind of model is Mitopark or CD157? 

We have added definitions and references for all animal models in the main results section. 

Are Tau models specific for Parkinson’s disease or for Alzheimer’s as well? 

We have addressed this with a clarifying statement in the Tau results of the appendix: “Whilst commonly associated with Alzheimer’s disease, mutations in the Tau protein have been linked to PD. Seven models contained a variant mutation of Tau...”